# False Positives in Brucellosis Serology: Wrong Bait and Wrong Pond?

**DOI:** 10.3390/tropicalmed8050274

**Published:** 2023-05-12

**Authors:** Borbála Bányász, József Antal, Béla Dénes

**Affiliations:** 1Department of Microbiology and Infectious Diseases, University of Veterinary Medicine Budapest, 1143 Budapest, Hungary; banyaszb@nebih.gov.hu; 2Laboratory of Immunology, Veterinary Diagnostic Directorate, National Food Chain Safety Office, 1143 Budapest, Hungary; 3Omixon Biocomputing Ltd., 1117 Budapest, Hungary; jozsef.antal@omixon.com

**Keywords:** *Brucella* spp., brucellosis, serology, false positive serologic results (FPSR), Gram-negative bacteria, lipopolysaccharide (LPS), smooth (S) colony types, rough (R) colony types, cell-free DNA (cfDNA), next generation sequencing

## Abstract

This review summarizes the status of resolving the problem of false positive serologic results (FPSR) in *Brucella* serology, compiles our knowledge on the molecular background of the problem, and highlights some prospects for its resolution. The molecular basis of the FPSRs is reviewed through analyzing the components of the cell wall of Gram-negative bacteria, especially the surface lipopolysaccharide (LPS) with details related to brucellae. After evaluating the efforts that have been made to solve target specificity problems of serologic tests, the following conclusions can be drawn: (i) resolving the FPSR problem requires a deeper understanding than we currently possess, both of *Brucella* immunology and of the current serology tests; (ii) the practical solutions will be as expensive as the related research; and (iii) the root cause of FPSRs is the application of the same type of antigen (S-type LPS) in the currently approved tests. Thus, new approaches are necessary to resolve the problems stemming from FPSR. Such approaches suggested by this paper are: (i) the application of antigens from R-type bacteria; or (ii) the further development of specific brucellin-based skin tests; or (iii) the application of microbial cell-free DNA as analyte, whose approach is detailed in this paper.

## 1. Introduction

Human and animal brucellosis and its health and economic consequences have been known for millennia. The pathogens behind the diseases, the *Brucella* species, were first described by Bruce, along with the rapid development of the first serologic diagnostic probe detecting *Brucella* infection by Wright at the end of the 19th century [1,2,3]. Since then, the genus *Brucella* has expanded to more than 30 known species, including isolates from exotic hosts, such as cetaceans or the surface of human breast implants [4], demonstrating the physiological and genetic flexibility of the bacteria.

Besides the ability to exist in a wide variety of hosts, this flexibility provides environmental persistence (against extreme temperature, pH, and humidity) outside of any host [5], as well as the ability of intracellular localization within the host organism [6,7,8] for most *Brucella* species. By completing the picture with the mild toxicity of the *Brucella* endotoxin—three orders of magnitude lower than the respective *E. coli* molecule [9]—the difficulties relating to the recognition and diagnosis of brucellosis are subsequently compounded. Perhaps it is not a coincidence that the most widely applied methods for diagnosing brucellosis are serologic tests, among which include a modified version of the very first one mentioned above [10].

### Brucellosis Serology: Catching Fish with the Wrong Bait in the Wrong Pond?

Three parallel demands in brucellae serology have been the greatest challenge for researchers, test developers, and authorities alike: (i) early and (ii) unambiguous determination of *Brucella* infection and (iii) the differentiation of an actual infection from an immune reaction resulting from vaccination. The old dream of serologists is to catch the “Golden Fish”—the perfect antibody in the host serum—which fulfills those three wishes. The third “wish” will not be the detailed subject of this paper, although it is of equal or higher importance than the second one in cases when vaccination is allowed.

Considering the responsibility of defining formally authorized serology probes, and the economic consequences of a positive *Brucella* infection test result on members of the livestock supply chain, it is not surprising that the confirmatory probe of brucellosis defined by the WOAH (former OIE) is still the reliable yet cumbersome complement fixation test (CFT) [10,11], even though it is not applicable for all ruminant species, for example, lamas [12].

For decades, the testing laboratories could access so-called brucellae-specific enzyme-linked immunosorbent assay (ELISA) probes, avoiding the tediousness of the CFT. However, ELISA remains a screening test only, further confirmation of the results is required. However, as it will be demonstrated below, not all serum antibodies reacting with *Brucella* antigens applied in the probes have been produced against bacteria belonging to *Brucella* spp. [13,14]. This fact is evident in the binary classification parameters of the brucellae indirect ELISA (iELISA) tests, if false positive serologic results (FPSRs) are considered (manufacturer claims usually do not covers FPSRs), which are results with high diagnostic sensitivity (>98%) and poor diagnostic specificity (<60%), with 95% confidence [15].

The risk of false positivity of the widely used serologic ELISAs for the detection of brucellosis is very high, to the extent that some laboratories apply not only the CFT, but parallel with it, the classic rose Bengal (RBT) [16,17] or/and serum agglutination (SAT) tests [2,3,18,19] for confirmation [13,20]. The diagnostic sensitivity and specificity of the WOAH-recommended probes are presented in Table 1.

The factors behind the relatively high prevalence of FPSRs are: (i) the ambiguous determination of cut-off values [21] and/or (ii) the molecular characteristics of the antigen–antibody interactions utilized in the probes. The fact that false positives can also occur in the confirmation tests instead of excluding false positives suggests that: (i) the confirmation tests have a tendency to confirm the same false positive results as those provided by the ELISA test (it is mainly their substantially lower analytical sensitivity that proves virtually the opposite in some cases); (ii) the common root cause of false positivity independently from the setup of the tests, that is both in screening ELISA and in the confirmatory tests is the antigen applied; and (iii) there should be more cross-reacting Gram-negative bacterium species or other unknown organisms or molecular antigens than we generally consider as contributing in false positives [20].

The application of the very same antigen resulting in false positives and the likely presence of unidentified contributors in such results deepens the uncomfortable sense that the fishing party for brucellae-specific serum antibodies is trying to catch fish with the wrong bait, possibly in the wrong pond.

## 2. Discussion

### 2.1. The Wrong Bait

With a perfect bait one could have a good catch even in the muddiest of waters, and in the right pond, even an otherwise imperfect bait could prove to be useful. In the first case, finding something which could attract the desired fish requires knowledge regarding the tastes and habits of our target, that is, of the antigenic nature of the *Brucella* cells attracting the specific anti-*Brucella* antibodies.

#### Issues with the Antigenicity of the Brucella Cell Wall

The members of *Brucella* spp. as Gram-negative bacteria are characterized by their sandwich-structured cell envelopes composed of the lipopolysaccharide (LPS)-covered bacterial outer membrane and the inner cytoplasmic cell membrane with a thin peptidoglycan layer between them in the periplasmic space.

Almost all members of the genus *Brucella* follow this scheme; however, with serious biological, biochemical, and subsequent serological consequences, similarly to other Gram-negative bacteria [22], there are some mutant *Brucella* species lacking the vast outer lipopolysaccharide layer [23]. These mutants were identified as R (rough)-type *Brucellas* in contrast with the S (smooth)-type species, based on the visual characterization of bacterial colonies grown on solid media. The virulence of R-type mutants is radically weakened due to the mutations not detailed here [23], making some of them applicable as *Brucella* vaccines.

According to their low prevalence, their importance from the standpoint of brucellosis serology is generally low. However, their application as vaccines, while providing opportunity for monitoring vaccination, could simultaneously prove to be an obstacle in the serologic distinction between vaccinated and infected populations.

The prevalence of the S-type *Brucella* species is substantially higher, hence false positivity in *Brucella* serology is mainly contributed by S-types. Therefore, in the following chapters, the biochemical and antigenic character of smooth *Brucella* spp. is discussed.

In our discussion, the description of the cell envelope will proceed from the internal parts of a bacterium (cytoplasm) to the extracellular space. Biological roles, structures, and cellular mechanisms involved in the generation of cytoplasmic proteins will not be detailed, only their antigenic characteristics will be discussed below as applicable [24,25,26]. The cytoplasm is surrounded by the inner membrane, built as a double layer (bilayer) from phospholipids and directly covers the cytoplasmic space, and is thus also referred to as the cytoplasmic membrane. There are various membrane proteins wedged into the inner membrane but, similarly to the protein content of the cytoplasm, only their antigenic characteristics will be considered [25]. At the outer side of the inner membrane, we find the periplasm: a structured, gel-like space with a peptidoglycan membrane distinctly different from the peptidoglycan membrane of Gram-positive bacteria, as it is not multilayered and is thus substantially thinner and more fragile. There are proteins dissolved in the periplasmic fluid, including enzymes (hydrolases, antibiotic-degrading enzymes, etc.), heavy metal neutralizers, carrier proteins, and bacterial toxin subunits. As in the case of cytoplasmic and inner membrane proteins, only their antigenic characteristics will be detailed as relevant to this paper [27].

The outermost part of the envelope is the cell wall composed of a phospholipid membrane monolayer and an associated lipopolysaccharide (LPS) layer. The outer membrane also anchors the periplasmic peptidoglycan layer by specific proteins: the outer-membrane proteins (OMPs). Some OMPs belong to the porin protein family providing molecular communication across the cell wall with their typical β-barrel (a tube-like structure of antiparallel β-folds) structure [28,29,30]. A detailed presentation of OMPs and their potential in *Brucella* serology will be provided below. Recent studies have proven that LPS is inhomogeneous: a mixture of full-length polysaccharide chains (S LPS) and truncated forms (R LPS) clustered around OMPs [22,31].

The asymmetric composition of the outer membrane can lead to the extremely increased hydrophobicity of the bacterium particle in all cases where the LPS becomes thinner due to mutations, causing the tendency for spontaneous congealing—similarly in the case of rough mutations.

The long (built from dozens of monosaccharides) polysaccharide chains of the LPS form a protective fur-like layer, making it difficult for hydrophobic molecules to penetrate the outer membrane and enter the periplasm. This strong fur coat lures the host immune system during the early phase of the infection and protects the *Brucella* cells from monocyte phagocytosis later, as referred below.

The LPS consists of three main elements distinct in composition, structure, and function. These elements, in order, starting from the interior of the bacterium cell toward the extracellular space are: (i) the O-specific polysaccharide (OPS), (ii) the core oligosaccharide (COS), and (iii) the so-called lipid A. Figure 1 presents the physical assembly of the main components only, without the detailed biochemical and biophysical processes (enzyme reactions, biochemical pathways, cell trafficking routes, and their elements [32]), resulting in the final composition and structure.

Lipid A, similar to phospholipids, consists of a polar head and a hydrophobic base. It is the key feature enabling the building of the asymmetric membrane bilayer of the outer membrane (OM). It is a conservative structure among Gram-negative bacteria, as detailed in Figure 2 and in the short discussion following the figure.

Figure 3 and its discussion presents the core oligosaccharide (COS), which, as its name suggests, is a core that links together subunits of LPS with different structures and functions, connecting the lipid A and the O-specific polysaccharide. Molecular diversity of the COS is high among bacterial species, including the strains of species.

The outmost part of the LPS, the O-specific polysaccharide (OPS), detailed by Figure 4 and Figure 5, is also known as O-specific side chain or O antigen. The OPS is composed of a high number of repeating subunits built of 2–7 sugar components of various compositions. The pattern of subunit repetitions is characteristic of bacterial species, resulting in high antigen variability, which can serve as the basis for the serological grouping of Gram-negative bacteria. The number of sugar moieties (which is dramatically reduced in rough mutants) in the O-specific polysaccharide chain determines the morphology of the bacterial colony, i.e., smooth or rough colony types with and without lengthy OPS chains, respectively. LPS which is lacking OPS due to a low degree of polymerization could be referred to as lipooligosaccharide—LOS.

Lipid A is usually built from a bisphosphorylated disaccharide with a rather conservative structure (the hydrophilic head), acylated with various fatty acid chains with variable lengths and branching, providing the hydrophobic base that noncovalently sticks to the hydrophobic side of the phospholipid monolayer in the OM.

Although it seems that lipid A structures, being the most deeply buried part of the LPS and embedded into the OM, have minor importance from a serologic point of view, as main determinants of bacterial endotoxicity—especially the low endotoxicity of brucellae, which amplify the stealthing ability of the bacteria [9,33,34]—some discussion of their antigenicity is required.

Lipid A of brucellae is the most hydrophobic membrane anchor among Gram-negative bacteria. The typical length of fatty acid chains in *Brucella abortus* is 16–18 with some very long (28–30) substituents [35,36,37] in contrast with the typical C12–C14 chains of enterobacteria [33,38]. The six long fatty acid substituents make it difficult to release the endotoxin for *Brucella*, while *Y. enterocolitica* lipid A, with only four and substantially shorter fatty acid chains, is a good source of free endotoxins.

The intracellular lifestyle that S-type brucellae enjoy in the endoplasmic reticulum [34] can be supported both by the firm anchorage of the LPS through lipid A—a useful feature to prevent dissolution in a lysosome as it generally happens with R-type bacteria —and by the attenuated host immune response [7,8,34]. Although diverse explanations have been published on the exact role of the LPS in such attenuations, data demonstrating the existence of S LPS-MHC II complexes [34,39,40,41] suggest a contribution by the extremely hydrophobic lipid A.

The core oligosaccharide has branched chains with two structurally different subunits, as Figure 3 demonstrates on the structure of *Brucella* COS, which are: (i) the inner core (CI) and (ii) the outer core (CO). We should mention that this classification of the subunits could be obsolete in the light of recent studies. The composition of the CI consists of rare sugar moieties, such as the characteristic ketodeoxyoctonic acid (Kdo) and heptose sugars or—such as in the case of *Brucella*—D-quinovosamine (QuiN) [42].

The terminal Kdo attaches the COS to the lipid A through a ketosidic bond, is sensitive to weak acids, and provides a link to the outer core as well. In the CO, the number of hexose monomers is variable with six sugar moieties as a maximum. Depending on the number of monomers, the CO could contain further small branches—in the case of *Brucella*, through the side chain of a D-glucosamine. A terminal moiety of a CI (such as in the case of *Y. enterocolitica* serotype O:3 and *Brucella* spp.) or a CO (*E. coli* and *Salmonella* spp.) branch attaches the COS to the OPS [33,42,43,44,45,46].

Investigations of the so-called deep rough mutants, such as the D31m4 of *E. coli* [47], determined the minimum length of the full COS in viable Gram-negative bacteria: two heptoses, usually Kdo monomers, should be attached to lipid A [47]. Rare exceptions exist, as in the case of *Helicobacter pylori*, with only one Kdo linked to lipid A.

The diverse composition and the branched structure of COS could serve as a good basis for antigenic differentiation; however, utilizing this diversity is difficult for serologic applications in the case of S-type bacteria as the majority of COS remain cryptic under the OPS layer. The mentioned diversity of the OPS chain length and the inhomogeneous cell surface location of the different OPS molecules, that is the clustering of R LPS molecules around OMPs [22,31], increases the theoretical possibility of an immune response against COS motifs on the intact cells and serologic detection of antibodies involved in such a response.

The O-specific polysaccharide as the outer layer of the cell wall is the main antigen determinant of the LPS and provides a solid, yet relatively unreliable, basis for the serologic differentiation of Gram-negative bacteria species and strains [48]. The molecular and topological diversity of the OPS is theoretically high: more than five dozen kinds of sugar moieties were identified as constituents of the polymer in various numbers, proportions, and clusters, built into linear or branched chains, and linked to even non-sugary substituents [39,49].

More than 180 different O-serotypes of *E. coli* have been identified before the high throughput genotyping era until 2005 [49] due to this variability, and, paradoxically, the high molecular diversity results in a lower antigenic diversity in several cases [49]. *E. coli* O35 and *Salmonella enterica* O62 or *E. coli* O98 and *Yersinia enterocolitica* O11,24 or *E. coli* O8 and *Klebsiella pneumoniae* O5 and our current subject, *Brucella* spp., *E. coli* O157:H7 and *Y. enterocolitica* O9 have identical or nearly identical antigens; in the last case, identical enough to present false positive serologic results.

The O-specific polysaccharide of the *Y. enterocolitica* O9 and the brucellae (see Figure 4) is characteristically a homopolymer of 4,6-dideoxy-4-formamido-alpha- D-mannopyranose (alpha-D-Rha4NFo) sugar moieties, which are linked to each other through α(1→2) and/or α(1→3) bonds in different proportions. As is shown in Figure 5, the prevalence of α(1→2) and α(1→3) bonds are different in the strains of *Brucella* spp. with an average proportion of the α(1→3) links between 0 and 20 percent. The first saccharide monomer of the OPS always has a reducing end and is linked to the next sugar through an α(1→2) bond. The serologic importance of the terminus of the OPS is high [50].

Principally, the serologic differentiation of Gram-negative bacterial strains is based on antibody recognition of repeating oligomeric saccharide motifs in the OPS rather than individual sugar moieties. As in the case of brucellae, whose serologic division was among the earlier classification attempts targeting bacterium species (carried out in 1932 by Wilson [51]), it may have been the very first one, preceding the serotyping of *E. coli* in 1944, or *Salmonella* spp. in 1935, during the extensive and pioneering efforts of Kaufman [52,53,54,55]. The early serotypes, M after *Brucella melitensis* and A after *Brucella abortus*, provided the basis for the later developed *Brucella* M and A antigen grouping and led to the recent stage of knowledge summarized in Figure 4 and Figure 5 [50,56,57].

*Brucella* OPS could carry the antigens A, M, C/Y, and C. According to recent knowledge [56], the A antigen represents an alpha-D-Rha4NFo moiety, which is linked between two alpha-D-Rha4NFo saccharides through α(1→2) bonds; or, suggesting a more simple determination: A antigen is two alpha-D-Rha4NFo moieties linked together by an α(1→2) bond (see Figure 4). The M antigen represents a cluster of four alpha-D-Rha4NFo moieties with a central link of an α(1→3) bond and the two saccharides on the termini of the cluster is linked by α(1→2) bonds; or, with a more simple determination: two A antigens linked together with an α(1→3) bond. As it is shown in Figure 5, at least one bacterium exists in the genus *Brucella* with uniform antigenicity: *B. suis* biovar 2 contains exclusively A antigens [58]. With similar rare exceptions, near the cap of the brucellae OPS, usually an α(1→3) bond can be found [50].

This molecular-based antigen division provides more sensitive typing since the determination of M and A dominant strains based on A/M ratios could identify strains between two extremis: the 100% A dominant *B. suis* biovar 2 and the *B. melitensis* 16M containing 21% of M [59].

The very existence of the C/Y and C antigens is still debated. Theoretically, they are characterized as overlapped A and M antigens with different proportions, that is, C/Y (as a common antigen of some members of the genus *Brucella* and *Y. enterocolitica* O:9) has more A than M and in C antigen A = M [60]—if we retain the fish and bait metaphor, they are baits for fish with the largest mouths—consequently with a lower target sensitivity.

Due to its chemical nature, which is abundantly composed of saccharides, and the rather narrow repertoire of sugar moieties occurring across bacterial families, from a serological point of view, the outermost polysaccharide component of the protective LPS layer seems to be overly uniform in the case of smooth Gram-negative bacteria; consequently, the application of isolated antigens with S LPS origin in serological tests almost automatically provides false positives. The baits used are evidently far from perfection.

### 2.2. The Wrong Pond

While the discussion above hints at what could be wrong with our old baits and has outlined some ideas on how to find better ones, all the fish populating the pond and the very muddiness of its “water” still requires some clarification.

#### 2.2.1. Muddy Waters: Issues with Cross-Reactive Species and Other Unidentified Agents

As our case study demonstrates, any immunoreactive agent in a host population providing antibodies reactive with the antigen in our tests—that is *Brucella* S LPS—could cause transient false positivity, consequently triggering *Brucella* eradication efforts. The pond was too muddy for the real source to be identified.

Antigenic substances overpopulate the serum with cross-reacting antibodies by provoking an immune reaction—that raises antibodies which interact with S LPS or its components—and create a low signal-to-noise ratio that consequently render the detection of real anti-brucellae antibodies impossible. In other words, the very muddiness of the pond creates so many ersatz Goldfish hungry for our imperfect baits that the real ones have no chance to reach them.

The cross-reactions with several Gram-negative bacteria, including *Y. enterocolitica* O:9, *E. coli* O:157, *S. urbana* O:30, *E. hermanii*, *Francisella tularensis*, *P. maltophilia* 555, *S. godesberg*, and *Vibrio cholerae*, have been well-documented since the advent of advanced serologic tests and is a concern to this day [15,20,56,61,62,63]. The high prevalence of the abovementioned, and other, still unidentified non-pathogenic bacteria strains causing mild or asymptomatic disease is a certain and unavoidable condition in our livestock.

Moreover, it should be considered that the muddiness of the pond originates from the “eagerness” of mammals’ advanced immune system *to react* and produce antibodies: to react *blindly* with almost any alien agent with bacterial origin or not. Thus, if we want to continue applying antigens with S LPS in our serologic test, during the interpretation of the test results, all possible sources of oligo- or polysaccharide antigens must be considered. This is a difficult, if not impossible, task, considering recent livestock-keeping practices and the far-from-reliable control of the international transportation of animals. Naturally, some research on non-bacterial antigens causing false positivity should be considered.

#### 2.2.2. Rivaling Schools of Fish: Time Lapse of Immune Reaction

The muddiness of the pond is not the only obstacle to catching a Golden Fish. Golden Fish have subpopulations and not all Golden Fish can fulfill all three wishes: our efforts could also be hampered by the occurrence of different antibody classes depending on the kinetics of the immune response.

*Brucella* infection first elicits a transient but strong humoral immune response [64,65]. The cellular response is prolonged, as it develops 3–4 weeks after infection [66] and can be detected for several years by a brucellosis skin test [67].

The volume of the recent knowledge of the stealthing mechanisms brucellae apply to modulate the host immune system in order to prevent inflammation at the site of infection, to avoid phagocytosis, to inhibit apoptosis of the host cells, and to inhibit cytokine response, etc., is substantially greater than the volume and the ambition of this paper; thus, we will focus on the antibody response only [7,8,34,38,68,69,70,71,72,73,74,75,76].

In acute brucellosis, the first and dominant immunoglobulin isotype is IgM [77], but on slow onset or if the initial IgM response could be undetectable, relapse of the disease may cause increased IgM levels [65]. The immune response subsequently switches to IgG isotype after 2–4 weeks [65].

Different types of serologic tests have different target sensitivity to the immunoglobulin isotypes; thus, the interpretation of the serologic results requires additional information about the temporal course of the infection on individual hosts. This type of information is usually missing/not provided.

The isotype switch—or the absence thereof—theoretically could be utilized for the differentiation of false positives from real ones [78,79]. However, our first requirement, that is the early detection of the infection, is highly compromised by the wait time for the abovementioned switch.

In the next part, we will present the many efforts that have been made to solve the target specificity problem, characteristic mainly of the ELISA methods, but existing even in the case of confirmation tests (see above).

### 2.3. Finding the Perfect Bait and/or Trying a Better Pond?

Since the first recognition of the ambiguity of the *Brucella* serologic tests to the point of revealing the molecular and microbiological background of that ambiguity, several efforts have been made to make the test free from FPSR or at least less ambiguous. At this stage of pursuing the Golden Fish, the challenge is to distinguish between the benefits of changing baits and choosing a better pond, that is, using non-LPS or LPS elements as the antigen and clarifying the muddiness caused by sugary antigens.

Theoretically, we can approach the problem from several directions: (i) the application of non-LPS or LPS elements (killed cells are considered as using LPS) as antigen in more recent test set ups; (ii) using non-S LPS or S LPS elements as antigen in more recent test set ups; and (iii) the use and development of techniques entirely different from traditional serology methods, that is ELISA, CFT, RBT, and SAT. A summary of accomplishments in this domain of scientific inquiry and practical achievements are shown in Table 2.

#### 2.3.1. Finding the Perfect Bait

While remaining in the sugary pond of oligo- or polysaccharide antigens, alternative approaches could be explored for finding better baits, such as: (i) using R LPS or R LPS elements or using killed rough-type bacteria, or (ii) applying artificial oligosaccharides shaped to avoid cross-reactions.

Applying R LPS or R LPS elements has the advantage of cheap and comparably simple realization (isolation from rough strains) and the high specificity proven in skin test.

The synthetic approach has higher costs and requires a deep knowledge of the molecular nature of the LPS and the availability of synthesis methods that make the production of the designed antigen possible. It may not be a simple coincidence that such set-ups were realized only in the last few decades (see Table 2). Theoretically, even rough saccharide motifs could be synthetized.

#### 2.3.2. Trying a Better Pond

Leaving the sugary pond as suggested means the application of other parts of the bacteria (protein(s) or peptide(s) (possibly peptide analogues)) with *Brucella* origin as baits.

By origin, such proteins could be isolated or recombinant ones (thanks to the biotechnology revolution) and, in the case of peptides and peptide analogues, synthesized. Isolation usually preserves the intact structure of the protein and is an inexpensive source of substances (except for membrane proteins, in which case, neither the low costs nor intact structures are guaranteed). The main risk of isolation is the remaining LPS impurities, whose total removal requires costly purification methods. Application of recombinant proteins or peptides reduces that risk to zero, but particularly in the case of membrane proteins, shaping and maintaining the native structures could be difficult if not impossible.

Cytoplasmic, periplasmic, inner, or outer membrane proteins or proteins of full cell isolates are good antigen candidates in theory and were tried in real test set-ups as outlined in Table 2. Upon applying periplasmic proteins, there is always a concern caused by the presence of periplasmic peptidoglycan layer, whose interactions with the periplasmic and trafficking proteins and the role of the layer in the maintenance of the proteins linked to it are still not satisfactorily understood.

This protein approach offers perfect baits and clear ponds in one solution: proteins and their respective peptides are more specific to bacterial species and even to strains, thus the occurrence of concurrent antigens has a lower probability (a clear pond), and the host immune response proceeds with higher specificity (a perfect bait). Investigations thus far suggest that the application of individual proteins provides no satisfactory solution for the FPSR, and that only a panel of *Brucella* proteins could solve the problem.

The requirement of early serologic detection could be compromised with the application of proteins of non-OM origin, since antigen presentation is a later result of the immune response when the invading cells are processed. Theoretically, using OM-originated proteins is a better solution, since a careful selection of protein surfaces or domains could provide targets for early recognition and reaction, especially in the light of our recent knowledge [22,31].

### 2.4. Stop Fishing and Start Hunting

The third approach simply leaves the whole fishing problem unresolved, choosing alternative methods to comply with all the three requirements stated in the first part of this paper. Using redeveloped or improved forms of existing methods such as skin tests (likely with newly developed antigens) or the application of blood serum or plasma samples used in serology but analyzed with non-serologic methods (such as cell-free DNA decoded by next-generation sequencing (NGS) or detected by digital PCR) simultaneously means leaving the dream of catching the Golden Fish and starting the hunt for the “*Miraculous Deer*”.

Skin tests utilize LPS-free fractions isolated from a rough *Brucella* strain (“brucellin”) and target the cellular immune response. This approach does not provide early detection and has some additional disadvantages. However, other than the longer response time to infection due to the reaction mechanism, (i.e., type IV hypersensitivity of the cellular immune response [101]), all other drawbacks, such as the lack of standardization, low throughput to handle a large amount of livestock, the difficulty of handling the extremely hydrophobic substances, and interpretation difficulties, etc., have a chance of being resolved and the method could eventually be standardized. The trophy of this hunt might be a relatively inexpensive and easily implementable test with proven high specificity.

Detection of cell-free DNA in blood plasma (or in serum, urine, or milk) derived from damaged cells leaves the immunological approach of brucellosis diagnosis trailing behind.

Although Mandel and Mëtais identified the cfDNA as early as 1948 [102], the concept of its use as analyte, applied in minimally invasive or non-invasive, rapid, sensitive, and accurate diagnostic methods of various diseases, remained mostly unrecognized until the last decade of the 20th century [103]. Since then, especially in the past few years, detection of cfDNA has become the proven diagnosis of various cancerous diseases (“liquid biopsies”) [104,105,106,107,108] and non-cancerous disorders such as diabetes [109], stroke [110], systemic lupus erythematosus [111], trauma [112,113], rheumatoid arthritis [114], inflammation [115,116], allograft rejection [117,118,119,120], and other infections [87,88,89,90,91,92,93,94,95,96,97,98,99,100]. As a minimally invasive method, cfDNA testing is widely used [121] in the prenatal determination of genetic disorders as an alternative to the riskier amniocentesis test.

The diagnostic potential of cfDNA as analyte in the detection and identification of infections is rather narrowly utilized, as the sporadic literature (mostly from the last decade) demonstrates. Of the hundreds of cfDNA papers, only a few dozen have been focused on infections. However, parallel with the step-by-step removal of drawbacks of isolation and detection technologies, which will be discussed below, the field has been gradually expanding and even encompasses our very subject: cfDNA detection was applied for the assessment of human brucellosis [87]. Additionally, this pioneering effort reveals to us the “Miraculous Deer”.

*Brucella* tests, based on cfDNA detection, can fulfill all three original requirements we stated, since, theoretically: (i) it works at any stage of the infection; (ii) it provides unambiguous determination of *Brucella* strains; and (iii) only the DNA from the active infection is detectable.

Considering the stealthy lifestyle of the *Brucella* species, the theoretical possibility of recognizing the pathogen at any stage of the infection highlights the advantage of the method: cfDNA is present in traceable levels in the early phases of the infection (preceding even the immune reaction, a prerequisite for serological detectability) and the presence of DNA derived from bacteria hidden in macrophages can also be detected at the later stages of infection.

Identification of *Brucella* strains and of non-*Brucella* pathogens for false positive samples provides the avoidance of the unnecessary slaughtering of animals and/or the appropriate treatment of the identified infection, as well as the possible prevention of such infections. Although all *Brucella* species and biovars share more than 90% DNA homology with accurate cfDNA markers, gene polymorphism allows differentiation. Simultaneously revealing the presence of *Brucella* and of unknown pathogens or even of multiple pathogens during one cfDNA detection process is also possible. Theoretically, any microbial agent whose genome is already—even partially—deciphered and aligned to curated databases could be identified through cfDNA.

The fulfillment of the third requirement is especially important in the case of vaccinated livestock, where real positive serologic results could be unambiguously confirmed as the consequence of vaccination and not of an active infection, without sacrificing animals for autopsy.

Despite the obvious advantages stated above, serious drawbacks delayed the wider spread of cfDNA detection in microbial diagnosis in general, with even wider lags in its veterinary application. These drawbacks stemmed partially from the nature of the analyte [122,123] and chiefly from difficulties encountered by the DNA identifying technology.

#### 2.4.1. cfDNA as Analyte

The main difficulties to be addressed by a diagnostic test of cfDNA samples are: (i) the low absolute concentration of the analyte; (ii) an inherently low signal-to-noise ratio, that is, its relative amount in comparison to the abundance of similar molecular species in the sample specimens (both are challenges for the detection technology); (iii) the fragmented nature of the cfDNA, resulting in a wide molecular weight range which poses an obstacle to molecular identification; (iv) the limited stability of the DNA molecule; and lastly, (v) the nature of the specimens themselves from which the analyte can be isolated. This lack of stability makes it difficult to ensure a DNase-free environment throughout the sampling workflow, challenging the processes of sample isolation, separation, and conservation technology [124,125,126,127,128,129,130,131].

The application of a polymerase chain reaction (PCR) provides a solution for drawback (i) through (ii), since highly specific amplification of the targeted cfDNA increases both the concentration the analyte and the signal-to-noise ratio. Moreover, with the PCR fragment assembly technique, DNA amplicons with appropriate detectable chain lengths could be produced, or fragments could be labelled to sensitive optical detection.

For the drawbacks related to the DNA stability, a wider and wider clinical application of the PCR technology already incited resolutions: a broad range of validated devices for sample management and preanalytical DNA stabilization are commercially available—at a competitive price [132].

#### 2.4.2. Identification of cfDNA

As outlined above, PCR technology alone resolves many drawbacks originating from the nature of the analyte, providing adequate amounts of specific DNA for analysis, and the appropriate sample management has also been resolved by the application of the well-known titer plate systems. However, the challenges related to DNA identification are increasing as diagnostic requirements become more rigorous.

In the simplest approach, which fulfills all the requirements we set, that is, the unambiguous determination of the presence or absence of an active *Brucella* infection and the identification of the *Brucella* strain causing said infection, can be accomplished in a single run of an electrophoretic separation of the DNA fragments [133,134] produced by a properly designed multiplex PCR. The current DNA electrophoresis systems also provide parallel separations and multiple—although with a limited number—sample processing within one run. For a low number of samples, this approach theoretically represents the shortest response time and the lowest cost of resolving false positive serologic results via cfDNA detection since this approach requires inexpensive instrumentation and only minimal genetic information with regard to the *Brucella* strains; consequently, no computing capacity other than that which is already built into a gel documentation system is applicable. 

Higher throughput and sensitivity can be reached by using capillary electrophoresis (e.g., genetic analyzers, which also provide the adequate fragment computation capacity) [135,136] in combination with advanced DNA barcoding [137] and fluorescence labelling [138] methods—naturally at substantially higher costs per sample.

The application of quantitative PCR (qPCR) methods [97] could provide higher throughput (up to the number of the wells of the PCR plate) and an even faster response. However, the costs of such an approach are substantially higher on account of both the required probes and the instrumentation itself. Moreover, the number of the available detection channels of a qPCR instrument restricts the multiplex sequence detection.

However, if there are no *Brucella* strains detected, as in the case of FPSR, the question remains open: what kind of pathogen provoked the immune response leading to the FPSR?

While the cfDNA can provide an answer, the identification of the non-*Brucella* pathogens significantly elevates both the costs and response time of the analysis. Fortunately, a medium sample throughput can be maintained with recent technologies. Building up even a multiplexed PCR primer system that can selectively amplify cfDNA from the set of possible infectious agents is not a difficult task. However, identification of the amplicons even in the case of a limited target set is impossible without high throughput DNA sequencing, i.e., the application of some next-generation sequencing technology. The recent state-of-the-art of the NGS requires costly instruments and preparation chemistry and the typical response time (including the sample preparation, PCR, and sequencing library build up) is more than 24 h. To balance this out, the high throughput NGS set ups, available with current technologies, could offer an affordable cost/sample ratio. Breakthrough technologies such as the nanopore DNA sequencing [139,140,141,142] could substantially reduce both the instrumentation cost (by a magnitude) and the response time at the cost of a relatively reduced sample throughput.

Although NGS results in an enormous volume of DNA sequence data requiring substantial computation power and appropriate algorithms to resolve the DNA code and identify the origin of the infectious agent, the computational task (thus costs and speed) for such identification is minuscule in comparison to the computational burden of HLA genotyping. HLA genotyping is based on a continuously growing number of >20,000 alleles in a curated database, and the computational challenges had been answered for almost a decade. For example, the Karius test applied by Degner et al. [87] for the identification of human brucellosis from cfDNA in plasma, offers the identification of >1500 organisms using as a database one order of magnitude smaller than the database used for HLA typing.

However, since the Karius test is based on NGS, both the sequencing and the DNA identification were provided as laboratory services—a solution which could be applicable in the human health systems, but hardly used in veterinary practice. 

In light of the experiences from the last decade, the current state of cfDNA identification does not represent state-of-the-art infection diagnostics, but is still too far advanced to be in common use, as demonstrated with the debate around the clinical utility of the approach used in the Karius test [98,99,100]. However, with the continuous development of NGS technologies, with the application of less expensive new methods, such as nanopore sequencing [141], and with the growing use of NGS applications, the costs will be substantially reduced and, eventually, the cost/benefit ratio could reach a threshold to allow cfDNA detection to be remunerative even in veterinary practice. This is especially impactful in the case of brucellosis, considering the economic effects of FPSRs.

Thus, the hunt is over.

## 3. Conclusions

This paper summarizes the status of the resolution of the general FPSR problem in *Brucella* serology based on our knowledge on the molecular background of the problem and highlights some prospective avenues for solutions. Due to the nature and ambitions of the paper as a general overview, only a high-level summary of the literature is provided. We attempt to illuminate the problem using a humble fishing metaphor which suggests the following conclusions: (i) fully resolving the FPSR problem, exclusively with the widely used serology tests, requires a deeper understanding both of *Brucella* immunology and of the serology tests than that which we currently possess, and (ii) the solutions themselves will be as expensive as the corresponding research, since the root cause of the FPSRs is the application of the same type of antigen (the S-type LPS) in currently approved tests. Thus, the new approaches to solving problems stemming from FPSR are necessary. Such approaches may include: (i) the application of R-type LPS as antigen, or (ii) the further development of highly specific skin tests based on the protein mixture isolated from R-type Brucella, or (iii) the application of microbial cell-free DNA as analyte. The third solution, if considered, may even cause a shift in the current *Brucella* eradication paradigm.

Keep in mind the old proverb: “When the bait’s worth more than the fish, it’s time to stop fishing”.

## Figures and Tables

**Figure 1 tropicalmed-08-00274-f001:**
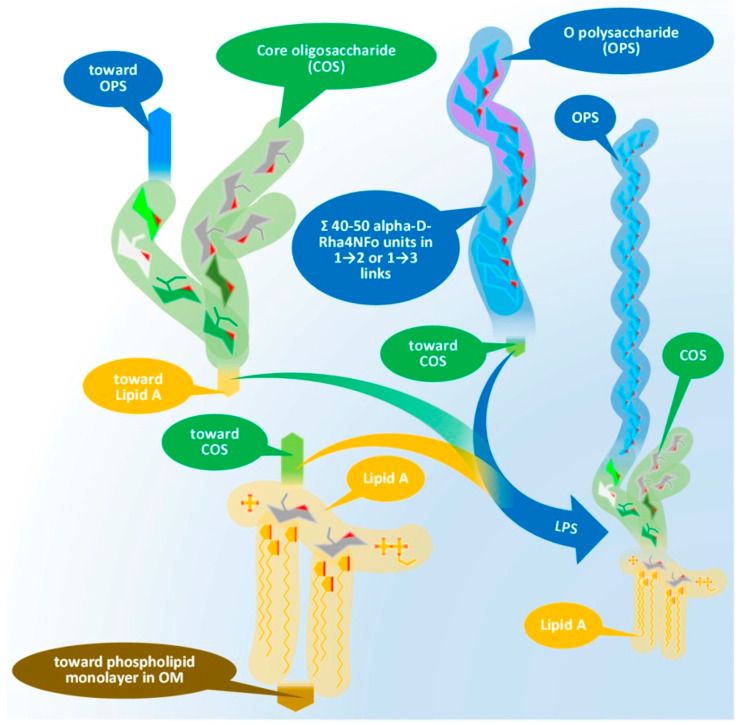
Physical assembly of LPS of Gram-negative bacteria demonstrated by the elements of *Brucella* LPS. Chemical compositions of the main components will be discussed later. LPS: lipopolysaccharide, OPS: O-specific polysaccharide, COS: core oligosaccharide, OM: outer membrane.

**Figure 2 tropicalmed-08-00274-f002:**
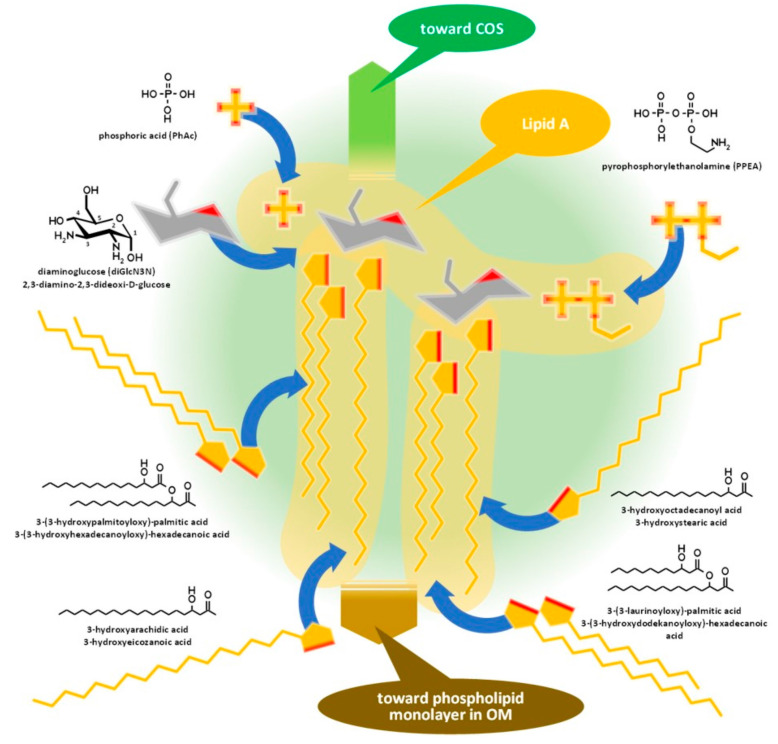
Assembly of lipid A of Gram-negative bacteria demonstrated by the composition of *Brucella* lipid A. COS: Core oligosaccharide, OM: outer membrane.

**Figure 3 tropicalmed-08-00274-f003:**
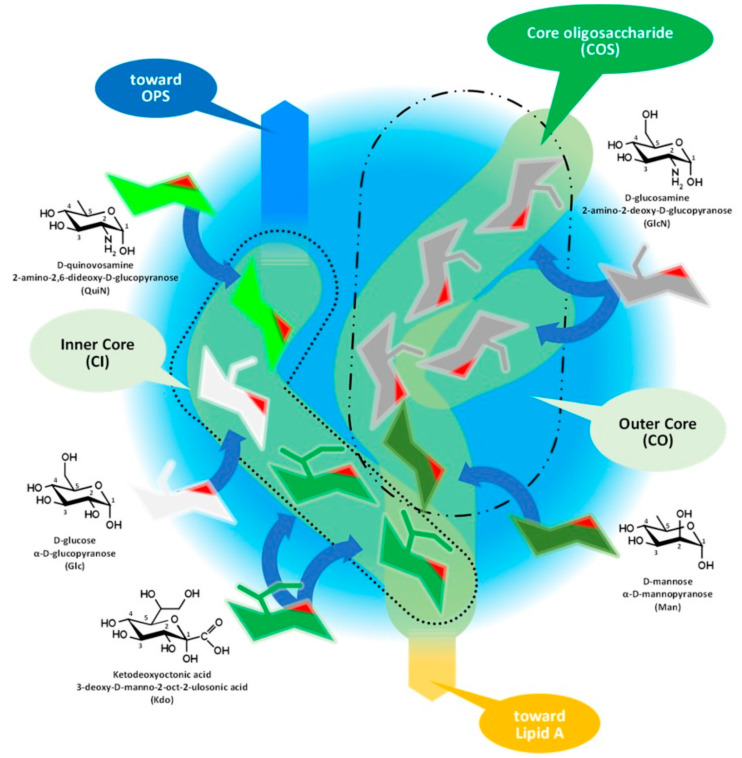
Assembly of core oligosaccharide of Gram-negative bacteria demonstrated by the composition of *Brucella* COS. OPS: O-specific polysaccharide.

**Figure 4 tropicalmed-08-00274-f004:**
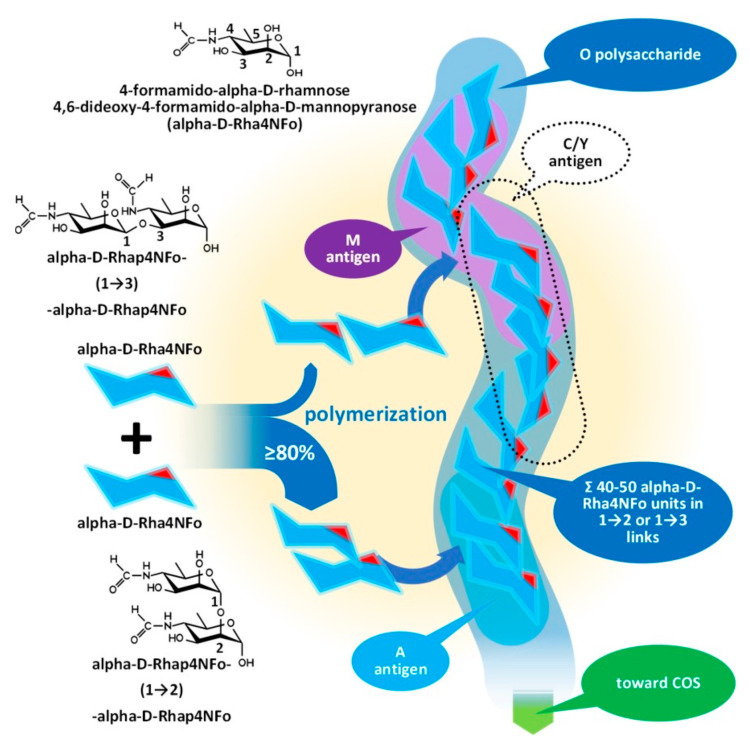
Assembly of the O-polysaccharide of Gram-negative bacteria demonstrated by the composition of *Brucella* OPS. COS: Core oligosaccharide.

**Figure 5 tropicalmed-08-00274-f005:**
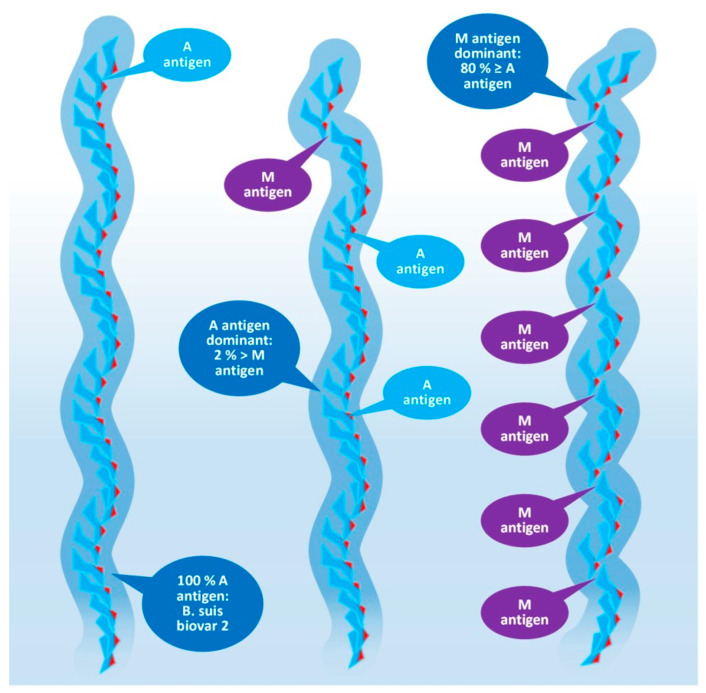
Composition of the O-polysaccharide of *Brucella* species with a different antigenic character.

**Table 1 tropicalmed-08-00274-t001:** Diagnostic sensitivity and specificity (%) of the WOAH-recommended probes.

Probe	Diagnostic Sensitivity (%)	Diagnostic Specificity (%)
RBT ^a^ [15]	100 (98.2–100)	86.4 (79.1–91.9)
CFT ^b^ [15]	100 (98.2–100)	94.4 (88.8–97.7)
iELISA ^c^ [15]	100 (98–100)	58.4 (49.2–67.1)

^a^: Rose Bengal test; ^b^: complement fixation test; ^c^: indirect ELISA.

**Table 2 tropicalmed-08-00274-t002:** Target samples to resolve the FPSR problems. Not all theoretic options are utilized in tests. NA: Not applicable. *: Recombinant protein. **: Recombinant chimera protein. ***: Fragmentation of R LPS. ****: Paper based.

Approach	Antigen/Target	Origin of Antigen	Target Molecule	Detection Technology	Reference
(i) Non-LPS or non-LPS elements	Other components of the bacterium cells (proteins)	Cytoplasmic	malate dehydrogenase *	NA	[24,25]
			superoxide dismutase *	iELISA	[25]
			bacterioferritin *	iELISA	[26]
		Periplasmic	protein 26 (BP26) *	iELISA	[27,80,81]
		Inner membrane	ABC transporter *	iELISA	[25]
		Outer membrane	Omp10 *	iELISA	[82]
			Omp16 *	iELISA	[82]
			Omp19 *	iELISA	[28]
			Omp25 *	iELISA	[28]
			Omp28 *	iELISA	[30]
			Omp19-25 **, Omp19-31 **, Omp25-31 **	iELISA	[83]
			OMP31*	iELISA	[28]
			OMP36 *	iELISA	[82]
			Omp2b *	iELISA	[29]
		Mixed origin	fusion protein of Omp16, Omp25, Omp31, Omp2b, and BP26 epitops (22 epitops) *	pELISA ****	[84]
(ii) Non-S LPS or non-S LPS elements	R LPS or R LPS element	Outer membrane isolate	NA	iELISA	[85]
		Processed *** outer membrane isolate	NA	NA	NA
		Killed cells	NA	iELISA	[86]
	Artificial oligosaccharides	Synthetic	free or BSA-conjugated oligosaccharides	iELISA	[50,56,59]
(iii) Alternative solutions	Brucellin	Killed cell isolates	NA	Skin test	[66]
	Cell-free DNA	Immune processed bacteria in blood plasma	DNA	Cell-free DNA NGS	[87,88,89,90,91,92,93,94,95,96,97,98,99,100]

## Data Availability

No new data were created or analyzed in this study. Data sharing is not applicable to this article.

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
