# Peer review of "False Positives in Brucellosis Serology: Wrong Bait and Wrong Pond?"

_tropicalmed, 2023, doi:10.3390/tropicalmed8050274_

Round 1

Reviewer 1 Report

This is a good comprehensive review of brucella serology trying to describe the reason behind false positivity and what tests could be used instead. It is also good that the authors mentioned the economic issue associated with the nucleic acid methods. A few comments should be addressed:

Major comments:

1. Line 52-52: You don't have to justify why you won't discuss the differentiation between infection and immunity due to vaccination by mentioning your occupation. The sentence isn't appropriate. You can just simply mention that this review will focus on the first two characteristics.

2. Lines 69-72: It would be helpful to have a small table comparing the different serologic tests in terms of their sensitivity and specificity percentages.

3. Although this review focuses on serology, there's no mentioning of the role of blood culture as a standard diagnostic method used by many laboratories. This could be briefly mentioned somewhere in the introduction or in the same paragraph as lines 69-72. Of note, one study found no correlation between serological results and blood culture at baseline. Hence, clinicians should not only focus on the result of serology alone or blood culture alone when diagnosing brucellosis (https://pubmed.ncbi.nlm.nih.gov/33531081/)

4. One potential reason behind false positivity beside the bacterial structural issue is the prevalence of false positivity in countries endemic for brucellosis, where people could have serologic titers >1:80 or even >1:160 without having symptoms that confirm the presence of an active infection. This should be briefly mentioned in the review when discussing reasons behind false positivity. There's an interesting study that looked at this issue using SAT as their serologic test: https://pubmed.ncbi.nlm.nih.gov/15188722/

Minor comments:

5. Line 41: "endotoxin [7] - 3 orders..." Looks like this was a typo. Please correct and clarify the sentence.

6. Lines 135-136 and 343-344: Better to cite the studies at the end of the sentence.

7. Line 423: There's a strange phrase that may have been inserted by mistake (Hiba! A hivatkozási forrás nem található)

Author Response

Dear Reviewer,

Thank you for the thorough revision of our manuscript. We also appreciate the valuable advice and critical remarks that were very useful. We agree with most of them, which helped us improve the manuscript's quality.

We corrected the manuscript and hope that our effort has reached its goal.

Below are the answers to the remarks in order following the review report:

Major comments:

  1. “Lines 52-52: You don't have to justify why you won't discuss the differentiation between infection and immunity due to vaccination by mentioning your occupation. The sentence isn't appropriate. You can just simply mention that this review will focus on the first two characteristics.

The sentence was corrected accordingly; see lines 52-52.

  1. “Lines 69-72: It would be helpful to have a small table comparing the different serologic tests in terms of their sensitivity and specificity percentages.”

The table and its references were inserted; see lines 73-76.

  1. “Although this review focuses on serology, there's no mentioning of the role of blood culture as a standard diagnostic method used by many laboratories. This could be briefly mentioned somewhere in the introduction or in the same paragraph as lines 69-72. Of note, one study found no correlation between serological results and blood culture at baseline. Hence, clinicians should not only focus on the result of serology alone or blood culture alone when diagnosing brucellosis (https://pubmed.ncbi.nlm.nih.gov/33531081/)”.

The authors appreciate the referee's suggestion. However, since the current WOAH manual states as not widely used and does not recommend blood culture as a veterinary diagnostic method, we could not find a place to refer to it among the serologic probes.

  1. ‘One potential reason behind false positivity beside the bacterial structural issue is the prevalence of false positivity in countries endemic for brucellosis, where people could have serologic titers >1:80 or even >1:160 without having symptoms that confirm the presence of an active infection. This should be briefly mentioned in the review when discussing reasons behind false positivity. There's an interesting study that looked at this issue using SAT as their serologic test: https://pubmed.ncbi.nlm.nih.gov/15188722/’

The suggestion was accepted, and the appropriate references are inserted; see lines 76-78.

Minor comments:

  1. Line 41: “endotoxin [7] – 3 orders…” Looks like this was a typo. Please correct and clarify the sentence.

Reference [7] was placed into a more precise position; see line 41.

  1. “Lines 135-136 and 343-344: Better to cite the studies at the end of the sentence.”

Corrected accordingly; see lines 141 and 143 and lines 348-349.

  1. “Line 423: There's a strange phrase that may have been inserted by mistake (Hiba! A hivatkozási forrás nem található).”

An invisible MS Word field code appeared in the .pdf version. The field code was detected and removed; See line 427.

Corrections by the authors:

Line 58: correction of the name of WOAH.

Lines 68-69: Wording replacement from ‘analytical’ to ‘diagnostic’ to resolve the ambiguity of probe properties.

Line 79: Insertion of Table 1 and the following paragraph, according to the suggestion, requires modification of the starting phrase of the sentence.

Lines 338-339: Corrected by a similar reason was mentioned in Comment 6.

Lines 367, 368, 381, 398: The table number was corrected due to the insertion of a new table by the suggestion of Reviewer 1.

Line 435: Mistyped reference was corrected.

Line 940-XXX: New literature references were inserted.

Sincerely yours,

Béla Dénes

Reviewer 2 Report

The authors of the publication presented the current state of knowledge on the phenomenon of false positive reactions in serological tests, organizing the current state of knowledge and presenting commonly known facts. As a reviewer, I would argue whether RBT is a confirmatory test. Conversely, RBT is a much simpler and less expensive test and is therefore used first. In case of a positive result, ELISA is used and in case of a positive result in this test, other tests, including CFT, are used at the end. In all guidelines for the diagnosis of brucellosis in animals, there is a rule that only the detection of the microorganism itself confirms the confirmation of brucellosis. By itself, DNA tests showing the presence of Brucella spp. genetic material are not proof of infection. At the same time, these guidelines indicate that tests for the presence of Yersinia enterocolitica O:9 should be performed, however, the authors also emphasize that there are other microorganisms that show false reactions. The only solution in the current situation is the elimination of animals suspected of infection, herd monitoring and, in some countries, vaccination, but it is difficult to distinguish the presence of antibodies after vaccination from antibodies formed during the course of infection. The authors in an extremely "poetic" way referred to goldfish, hunting or fishing, which makes the work even more interesting. The technical note concerns line 420 and a sentence fragment in Hungarian.

English is correct. The technical note concerns line 420 and a sentence fragment in Hungarian.

Author Response

Dear Reviewer,

Thank you for the thorough revision of our manuscript. We are glad you have a favorable opinion of our work and the manuscript. As you can see, we corrected the text. We hope that our effort has reached its goal.

Sincerely yours,

Béla Dénes

Round 2

Reviewer 1 Report

Excellent revision. Thanks to the authors for addressing the comments to improve their manuscript. I only have one minor comment: Please add the definitions of the abbreviations under Table 1. Other than this minor issue, I think the manuscript could be accepted.

Author Response

Dear Reviewer,

Thank you very much for your suggestion. As you recommended, we have inserted the definitions of the abbreviations under Table 1. 

Sincerely yours,

Béla Dénes